# Inner Rotation of Pickering Janus Emulsions

**DOI:** 10.3390/nano11123312

**Published:** 2021-12-06

**Authors:** Rajarshi Roy Raju, Joachim Koetz

**Affiliations:** Institut für Chemie, Universität Potsdam, Karl-Liebknecht-Strasse 24-25, D-14476 Potsdam, Germany; Raju_cep@yahoo.com

**Keywords:** Janus droplets, Pickering emulsions, magnetic manipulation, gold nanoparticles, magnetite nanoparticles

## Abstract

Janus droplets were prepared by vortex mixing of three non-mixable liquids, i.e., olive oil, silicone oil and water, in the presence of gold nanoparticles (AuNPs) in the aqueous phase and magnetite nanoparticles (MNPs) in the olive oil. The resulting Pickering emulsions were stabilized by a red-colored AuNP layer at the olive oil/water interface and MNPs at the oil/oil interface. The core–shell droplets can be stimulated by an external magnetic field. Surprisingly, an inner rotation of the silicon droplet is observed when MNPs are fixed at the inner silicon droplet interface. This is the first example of a controlled movement of the inner parts of complex double emulsions by magnetic manipulation via interfacially confined magnetic nanoparticles.

## 1. Introduction

Janus emulsions containing two non-mixable oil components dispersed in water have attracted great interest in different fields of application; they are superior to single emulsions because of the different interfaces and advanced morphological structures [1,2,3,4,5,6]. Different strategies for the preparation of Janus emulsions are already well established, e.g., by microfluidic processes [7,8] or one-step vortex mixing in a mini shaker [4,5,6,9,10]. Numerical analysis has shown that the angles at the contact line between three liquids can be calculated from experimentally obtained interfacial tensions [11]. The droplet topology is thermodynamically favored over a limited range of interfacial tensions < 5 mN/m [11]. To realize the conditions for equilibrium topology, surface tension active components, e.g., surfactants or polymers should be added. Friberg et al. have shown that in a ternary system containing olive oil (OO), silicone oil (SiO) and water the nonionic surfactant Tween 80 can fulfill these requirements very well [12]. Furthermore, it was shown that phospholipids [10], gelatin and chitosan [13] and thermo-responsive polymers [14] can reduce the interfacial tension in the ternary OO/SiO/water system in a similar way. Therefore, it becomes possible to produce Janus emulsions with pH- and temperature-responsive properties [14,15]. Recently, we showed that superparamagnetic magnetite nanoparticles (MNPs) can be successfully incorporated into olive oil [15]. This opens the door to introducing magnetic properties into Janus droplets. On applying an attractive stimulus, individual Janus droplets, as well as droplet clusters, start to rotate in an applied magnetic field [15]. Zentner et al. showed a controlled movement of complex double emulsions via interfacially confined magnetic nanoparticles [16], where amine-functionalized Fe_3_O_4_ nanoparticles selectively attached to one of the interfaces of double emulsions via covalent binding [17]. Ge et al. demonstrated the controlled group motion of anisotropic Janus droplets [9]. The continuous rotation of the Janus droplets as microreactors offered excellent performances in pollution adsorption and separation experiments [9]. Guo at al. prepared surfactant-free magnetic ionic liquid–water Janus droplets that could be stimulated by magnetic attraction [18]. The magnetic manipulation of Janus droplets of different sizes and shapes is still a challenge in developing stimuli-responsive colloid motors [19,20,21].

In this work, we produced completely engulfed core–shell Janus droplets stabilized by colloidal gold nanoparticles (AuNPs) in the outer shell and magnetite nanoparticles (MNPs) in the core shell. These “double” Pickering Janus emulsions combine the optical properties of the AuNP shell with the magnetic properties of the inner MNP shell. The red-colored Janus droplets can be stimulated by an applied magnetic field. Surprisingly, in the magnetically manipulated system only the inner droplet rotates. To the best of our knowledge this is the first example of a controlled rotation of the inner droplet in a core–shell Janus emulsion.

## 2. Materials and Methods

### 2.1. Materials

Silicone oil (SiO) with a viscosity of 10 mPa s, a density of 0.93356 kg/m^3^ and a refractive index (RI) of 1.39922, and olive oil (OO) with a density of 0.90865 kg/m^3^ and RI = 1.46689 were obtained from Sigma-Aldrich. Tetrachloroauric acid trihydrate (HAuCl_4_ 3H_2_O) was purchased from VWR. A low-molecular-weight chitosan with a degree of deacetylation of 81.2% was acquired from Sigma-Aldrich. Iron(III)chloride hexahydrate (FeCl_3_ 6H_2_O) was obtained from Fluka and iron sulfate heptahydrate (FeSO_4_ 7H_2_O) from Roth. For all experiments, purified water from a Milli-Q Reference A+ System (Millipore) was used.

### 2.2. Synthesis of Chitosan-Stabilized Gold Nanoparticles

A total of 0.05 wt.% chitosan was dispersed in a 0.1 mol acetic acid solution and heated to 80 °C for 10 min. Afterwards, 200 µL of HAuCl_4_·3H_2_O (5 mM) was added. The red color after 25 min indicates the nanoparticle formation. The resulting AuNPs with a mean particle size of about 18 ± 2 nm were used for the Janus emulsion preparation.

### 2.3. Synthesis of Magnetite Nanoparticles

Fe_3_O_4_ magnetic nanoparticles (MNPs) were prepared by mixing FeCl_3_ 6H_2_O (5.838 g; 0.0216 mol) and FeSO_4_ 7H_2_O (3.003 g; 0.0108 mol) in the ratio of 2:1 in a three-necked flask with 100 mL of water. Afterwards, 7.5 mL of ammonium hydroxide was added to the mixture and a black precipitate of MNPs was formed. The reaction was carried out at 85 °C for 25 min under inert conditions (co-precipitation method according to Liu et al. [22]). The nanoparticles were separated and washed several times with water and a 0.02 M NaCl solution. The purified MNPs were stored in a drying cabinet under vacuum. The resulting MNPs with a mean particle size of 13 ± 2 nm were dissolved in olive oil for the preparation of Janus emulsions.

### 2.4. Preparation of Janus Emulsions

At first, MNPs were dispersed in olive oil by ultrasound treatment for 10 min in an Elma Transsonic 460/H ultrasonic bath. The concentration of MNPs in olive oil was 1 mg/mL. Afterwards, olive oil containing the MNPs, water containing the AuNPs and silicone oil were mixed together in a 10 mL vial by shaking for 1 min in a mini shaker. The emulsion volume was 2 mL, with 100 µL of each oil component. The prepared emulsion was stored at room temperature for further measurements.

### 2.5. Methods

#### 2.5.1. Light Microscopy

Light microscopy images of the emulsions were obtained using a Leica DMLB microscope and videos were obtained with a Leica DFC295 live camera. A drop of the emulsion was placed on a glass slide (76 × 26 mm) and captured by varying the magnification. The images present a realistic picture of the drop topology in bulk emulsions.

#### 2.5.2. Cryo-Scanning Electron Microscopy (cryo-SEM)

Janus emulsions were plunged into nitrogen slush at atmospheric pressure, freeze-fractured at −180 °C, etched for 60 s at −98 °C and sputtered with platinum in a Gatan Alto 2500 cryo-preparation chamber. After transfer into the S-4800 cryo-SEM (Hitachi, Tokyo, Japan), the morphology of the emulsion droplets was evaluated at an acceleration voltage of 2.0 kV.

#### 2.5.3. Transmission Electron Microscopy (TEM)

AuNP- and MNP-containing solutions were dropped on carbon-coated copper grids. After solvent evaporation, the air-dried samples were examined in a JEOL JEM-1011 TEM at an acceleration voltage of 70 kV. The mean particle size was calculated from more than 150 nanoparticles.

#### 2.5.4. Magnetization Experiments

The experiments were performed with 4 neodymium cube magnets (10.0 × 10.0 × 10.0 mm^3^) with technical specification for magnetization grade N42 and a remanence of 1.3 tesla. An external setup allowed us to rotate the magnets around the glass slides under the light microscope by hand, clockwise and anti-clockwise.

## 3. Results and Discussion

AuNPs were synthesized in the presence of chitosan as a reducing and stabilizing agent in aqueous solution. The mean particle size of the resulting AuNPs was about 18 ± 2 nm, as seen in the TEM micrograph in Figure 1a. Separately synthesized MNPs with a mean particle size of 13 ± 2 nm showed superparamagnetic properties (compare with Figure 1b), as already shown by us earlier [15].

Initial experiments showed that the chitosan-stabilized AuNPs dispersed in the aqueous phase could successfully stabilize a Janus emulsion with olive and silicon oil. The resulting freshly prepared Janus droplets showed a core/shell morphology, where the silicon droplets were completely engulfed by the olive oil (Figure 2a). The individual droplets (compare with Figure 2b) are surrounded by a red-colored ring, which can be related to the AuNPs attached at the oil/water interface. The cryo-SEM micrograph in Figure 2c shows one individual large Janus droplet with a broken shell. Therefore, one can conclude that a Pickering Janus emulsion with AuNPs at the olive oil/water interface is formed in the absence of other reducing or stabilizing components.

Hydrophobic MNPs incorporated into the olive oil phase in the absence of AuNPs can turn to the oil/water interface or to the oil/oil interface. Recently, we showed that MNPs located at the oil/water interface drastically decrease the size of the Janus droplets [15]. Stimulation by an external magnetic field leads in this case to overall movement of individual Janus droplets and cluster aggregates, as shown by us earlier [15].

In the described system, a droplet-minimizing effect is missing. This is the first hint that the MNPs are not located at the oil/water interface, which is already occupied by the chitosan-stabilized AuNPs. In consequence, the resulting Janus emulsion (Figure 3) shows the typical core–shell structure, similar to the system discussed above in the presence of AuNPs alone, where silicon droplets are engulfed by the olive oil.

When an external magnetic field is applied to Janus emulsions with chitosan-stabilized AuNPs and MNPs, a surprising result is observed. By moving the magnet clockwise or anti-clockwise, a controlled movement of the inner droplet takes place, as can be seen in Figure 4 and the corresponding video in the Appendix A. Electromagnets were not used in these experiments for the sake of simplicity. The movement of the droplets was clearly affected by the gradient of the field, and the inner core of the Janus droplet moved in a linear translation with the movement of the magnetic field. Initial experiments showed a strong correlation between droplet movement and the rotating frequency of the magnet, and also the field strength.

This behavior can be explained only by the fact that the MNPs are located at the oil/oil interface. This is the first example of a “double” Pickering Janus emulsion with stimuli-responsive nanoparticles at the oil/oil interface and AuNPs at the oil/water interface, as shown schematically in Figure 5.

## 4. Conclusions

We prepared completely engulfed Janus droplets stabilized by chitosan-stabilized AuNPs at the oil/water interface and stimuli-responsive MNPs at the inner oil/oil interface. This new type of “double” Pickering emulsion offers the possibility of rotating the inner oil droplet by magnetic manipulation with an external magnet. To the best of our knowledge, this is the first example of selective movement of the inner droplet in Janus emulsions. Future experiments with these types of Janus emulsions will offer new application possibilities, e.g., for the construction of stimuli-responsive micro-lenses with rotating inner oil droplets of higher refractive index (according to [2]) or for magnetic hyperthermia.

## Figures and Tables

**Figure 1 nanomaterials-11-03312-f001:**
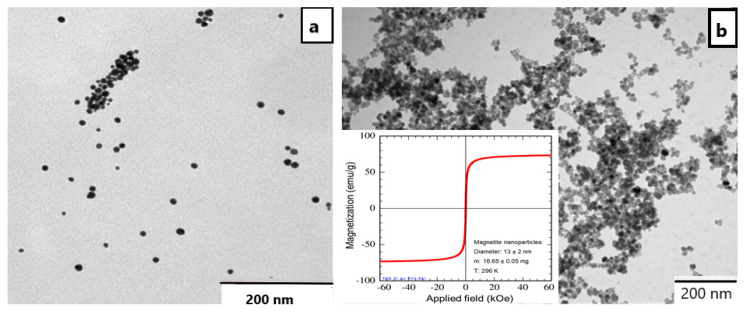
TEM micrographs of chitosan-stabilized AuNPs (**a**) and separately prepared MNPs with the corresponding magnetization curve (**b**).

**Figure 2 nanomaterials-11-03312-f002:**
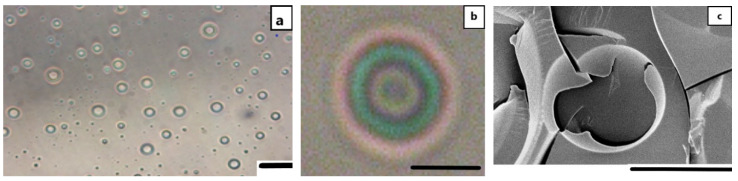
Light micrographs of a freshly prepared Janus emulsion with chitosan-stabilized AuNPs (scale bar: 20 μm) (**a**), one individual Janus droplet with a red-colored AuNP layer (scale bar: 5 μm) (**b**) and a cryo-SEM micrograph of a broken Janus droplet (scale bar: 5 μm) (**c**).

**Figure 3 nanomaterials-11-03312-f003:**
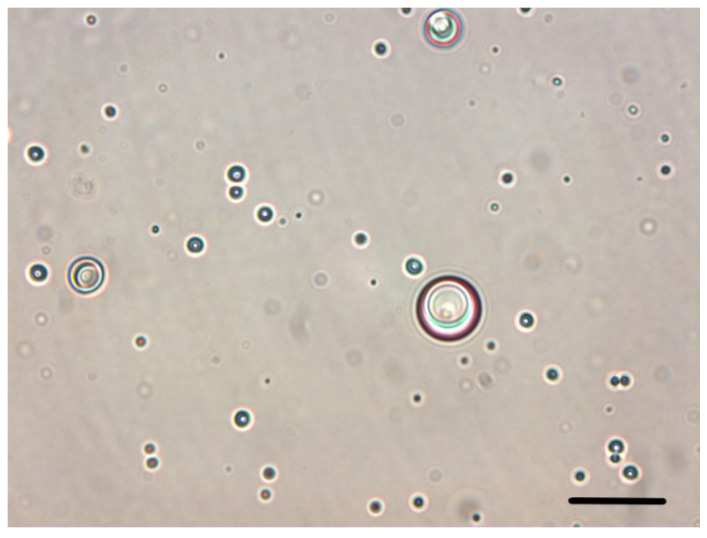
Light micrograph of a freshly prepared Janus emulsion with chitosan-stabilized AuNPs and MNPs (scale bar: 20 μm).

**Figure 4 nanomaterials-11-03312-f004:**
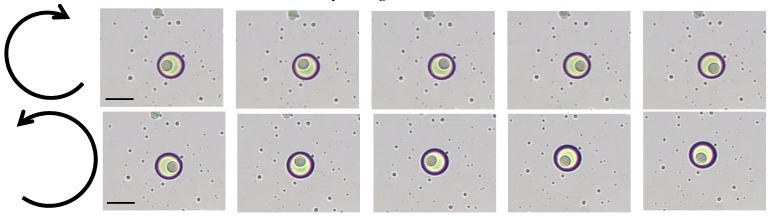
Rotation of the inner silicon droplet clockwise or anti-clockwise by moving an external magnet around the sample (scale bar: 20 µm).

**Figure 5 nanomaterials-11-03312-f005:**
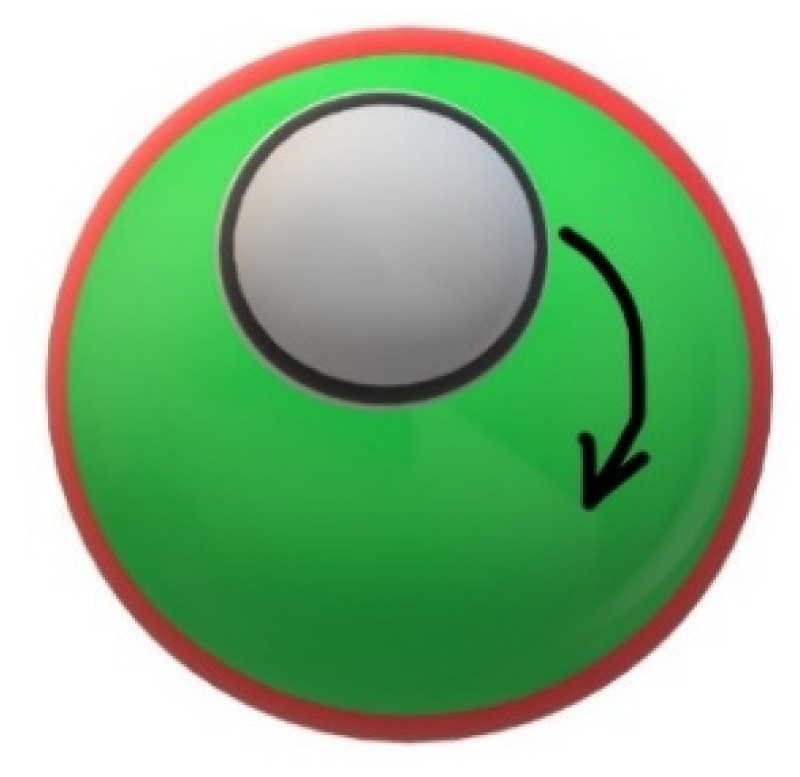
Schematic of the rotating Janus droplet with a black-marked MNP layer at the oil/oil interface and a red-marked AuNP layer at the oil/water interface.

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
