# Peer review of "Inner Rotation of Pickering Janus Emulsions"

_nanomaterials, 2021, doi:10.3390/nano11123312_

Round 1

Reviewer 1 Report

In this article, Roy Raju and Koetz have prepared emulsions with a Janus structure that confine gold and magnetite nanoparticles in different layers. Such droplets are sensible to the presence of magnetic fields and the inner droplet rotates.   This work presents innovative results with scientific and technological interests. The used methodology is suitable and is conducted properly. However, more studies of the magnetic excitations on the Janus emulsions are required.  The following points need to be addressed:    • What is exactly the used configuration of the magnetic field? Is applied this field with a rotating magnet or with an electromagnet?.  More information about the application of the magnetic field is needed.   • Can the droplets move in linear translations guided by a gradient of the magnetic field?.   • Are there a dependence of the rotating frequency and the magnetic field?   • Do irradiate heat the rotating droplets at higher rotation frequencies? It could be interesting for magnetic hyperthermia applications.  

Reviewer 2 Report

The work of Raju and Koetz reports the first evidence of a controlled movement of nanoparticles in Pickering Janus emulsions. The topic is of great interest and the results are surprising.

For this reason, I consider the work appropriate for the publication on Nanomaterials after minor modifications.

  • Page 2 line 59 and 60 in Kg/m3, “3“must be superscript.
  • Page 2 line 61 there is an error in the formula of tetrachloroauric acid.
  • The reference section needs to be adjusted.
